# A Novel Metabolic Risk Classification System Incorporating Body Fat, Waist Circumference, and Muscle Strength

**DOI:** 10.3390/jfmk10010072

**Published:** 2025-02-22

**Authors:** Carlos Raúl Robledo-Millán, María Regina Diaz-Domínguez, Ari Evelyn Castañeda-Ramírez, Efrén Quiñones-Lara, Sebastián Valencia-Marín, Ricardo Xopán Suárez-García, Nely Gisela López-Desiderio, Claudio Adrían Ramos-Cortés, Areli Marlene Gaytán Gómez, Juan Manuel Bello-López, Héctor Iván Saldívar-Cerón

**Affiliations:** 1Carrera de Médico Cirujano, Facultad de Estudios Superiores Iztacala, Universidad Nacional Autónoma de México, Tlalnepantla 54090, Mexico; carlosraulromi@gmail.com (C.R.R.-M.); diaz.regina@gmail.com (M.R.D.-D.); aricasra206@gmail.com (A.E.C.-R.); efrenquinolara@gmail.com (E.Q.-L.); sebasvalm27@gmail.com (S.V.-M.); ricardoxopan@gmail.com (R.X.S.-G.); nlopezd2300@alumno.ipn.mx (N.G.L.-D.); claussramosone@gmail.com (C.A.R.-C.); marlenegaytangmz@gmail.com (A.M.G.G.); 2Laboratorio 14, Unidad de Biomedicina (UBIMED), Facultad de Estudios Superiores Iztacala, Universidad Nacional Autónoma de México, Tlalnepantla 54090, Mexico; 3Laboratorio de Medicina de Conservación, Escuela Superior de Medicina, Instituto Politécnico Nacional, Mexico City 11340, Mexico; 4Hospital Juárez de México, Mexico City 07760, Mexico; juanmanuelbello81@hotmail.com

**Keywords:** metabolic risk, obesity, grip strength, metabolic syndrome, risk stratification, public health, body composition

## Abstract

**Background:** As metabolic diseases continue to rise globally, there is a growing need to improve risk assessment strategies beyond traditional measures such as BMI and waist circumference, which may fail to identify individuals at risk. This study develops and validates a novel metabolic risk classification system that incorporates body fat percentage (%BF), waist circumference (WC), and grip strength (GS) in Mexican adults. It aims to improve risk stratification and evaluate the association with metabolic syndrome. **Methods**: This cross-sectional study involved 300 young adults (18–22 years) from a university in Mexico City, utilizing body composition (%BF) and anthropometric measures (WC, GS) to categorize them into four risk groups: protective, low risk, increased risk, and high risk. A retrospective cohort of 166 adults (18–65 years) with complete clinical records was used for validation. **Results**: The inclusion of GS in the risk assessment significantly shifted the distribution in the young adult cohort, reducing the “no risk” category (15.5% males, 11.6% females) and expanding the higher-risk categories (70.2% males, 69% females). Metabolic parameters such as fasting glucose, triglycerides, HDL cholesterol, and blood pressure worsened progressively across the risk categories (*p* < 0.001). The high-risk group exhibited a markedly increased odds ratio for metabolic syndrome at 28.23 (10.83–73.6, *p* < 0.001), with no cases in the protective and low-risk groups. **Conclusions**: Integrating grip strength with %BF and WC into a risk classification system substantially enhances metabolic risk stratification, identifies at-risk individuals not previously detected, and confirms a protective group. This validated system provides a robust tool for early detection and targeted interventions, improving public health outcomes in metabolic health.

## 1. Introduction

The increasing prevalence of metabolic diseases, such as type 2 diabetes mellitus and cardiovascular disorders, underscores the need to refine risk assessment methodologies. Traditional indicators like BMI and waist circumference provide limited insight into metabolic health, often overlooking individuals with hidden risk factors [1]. Traditionally, assessments have predominantly relied on body mass index (BMI) and waist circumference (WC), which are widely used due to their simplicity and correlation with metabolic risk factors [2]. However, these conventional indicators often fail to capture the nuanced complexity of metabolic phenotypes, particularly in populations with distinctive characteristics, such as young Mexican adults, who may present unique risk factors [3,4,5].

Recent studies have proposed a novel phenotyping system for people with obesity that combines body fat percentage (%BF) and WC to create a matrix of nine phenotypes, classified into five cardiometabolic risk categories ranging from “no risk” to “very high risk” [6]. This system addresses the limitations of BMI by providing a more accurate assessment of both adiposity and its distribution, which are critical in predicting metabolic complications [7]. The utility of this system has been validated in large cohorts, demonstrating its ability to reclassify a significant portion of individuals into higher risk categories that were previously undetected by existing methods, thus offering a more precise tool for clinical practice and research [8].

In recent years, muscle strength has gained attention as a significant factor in metabolic regulation and the prevention of sarcopenia, a condition marked by the loss of muscle mass and function [9,10]. Sarcopenia is traditionally associated with aging, but it is increasingly recognized as relevant in younger populations, where reduced muscle strength is linked to heightened risks of metabolic diseases and increased mortality [11,12]. Despite this growing recognition, the potential of muscle strength as an integral part of metabolic risk assessment remains underexplored, particularly in younger demographics.

Incorporating muscle strength into metabolic risk assessments, in addition to %BF and WC, offers the potential for a more comprehensive understanding of metabolic phenotypes [13]. Reduced muscle strength, even in the presence of normal %BF and WC, may indicate a higher metabolic risk, which is often overlooked by current methods [14]. These findings underscore the value of a combined assessment of %BF, WC, and muscle strength to more accurately identify at-risk individuals and guide the development of targeted, effective interventions [15].

This study aims to develop a novel classification system that integrates %BF, WC, and GS and validate it in a population of Mexican adults aged 18–65 years. By analyzing its association with key metabolic parameters—fasting glucose, triglycerides, HDL cholesterol, and blood pressure—this study seeks to establish the system’s robustness and potential for broader adoption in diverse populations. This approach not only advances the precision of metabolic risk assessments but also supports the development of targeted strategies for early intervention and prevention.

## 2. Materials and Methods

### 2.1. Subjects

Participants were young adults aged 18–22 years, recruited from the metropolitan area of Mexico City. The study population included students enrolled in the medicine program at the Faculty of Higher Studies Iztacala (FESI), National Autonomous University of Mexico (UNAM). A non-random sampling method was employed to achieve broad representation within the student population. Recruitment was conducted via in-class announcements and extracurricular activities, resulting in a participation rate of approximately 90%. Data collection occurred between April and May 2024.

Inclusion criteria consisted of being actively enrolled in the medicine program at FESI, UNAM; being within the specified age range; voluntarily agreeing to participate by providing written informed consent; and having no self-reported history of metabolic disorders such as diabetes, hypertension, or dyslipidemia. Exclusion criteria included pregnancy or lactation; medical conditions that could affect body composition or muscle strength, such as neuromuscular disorders or severe musculoskeletal injuries, the use of medications known to influence metabolic parameters or muscle function; the presence of implanted electronic devices such as pacemakers due to the use of bioelectrical impedance analysis (BIA); or incomplete anthropometric or biochemical data.

For the validation phase, a retrospective cohort of 166 clinical records of Mexican adults aged 18–65 years was analyzed. Inclusion criteria required complete records with anthropometric data, bioimpedance analysis, grip strength measurements, and laboratory data for fasting glucose, blood pressure, triglycerides, and HDL cholesterol. Clinical records were collected between January 2023 and October 2024.

### 2.2. Study Design

This cross-sectional study received approval from the Ethics Committee of FESI, UNAM (CE/FESI/032024/1698). All participants provided written informed consent prior to participation. To ensure confidentiality, all data were anonymized and securely stored. Each participant underwent a comprehensive medical history review and a detailed physical examination to assess metabolic risk factors.

### 2.3. Anthropometric and Body Composition Measurements

Height was measured using a SECA mechanical stadiometer (SECA GmbH & Co., KG, Hamburg, Germany). Body weight and composition were assessed using a multifrequency bioelectrical impedance analysis (BIA) scale (InBody 120, Biospace Co., Seoul, Republic of Korea), with participants being measured while barefoot and minimally clothed. Body mass index (BMI) was calculated as weight (kg) divided by height squared (m^2^). Waist circumference (WC) was measured at the midpoint between the last rib and the iliac crest using a non-elastic SECA ergonomic measuring tape (SECA GmbH & Co., KG, Hamburg, Germany) with a precision of 0.1 cm.

All measurements were conducted by trained personnel certified in anthropometric assessment following standardized protocols to ensure consistency. Evaluators received formal training based on the guidelines established by the International Society for the Advancement of Kinanthropometry (ISAK). To assess measurement reliability, the inter-tester and intra-tester technical error of measurements (TEMs) was calculated prior to data collection. The intra-tester TEM for height and waist circumference was below 1.0%, while inter-tester TEM remained under 1.5%, both within acceptable ranges for anthropometric assessments [16].

### 2.4. Grip Strength Measurement

Grip strength (GS) was evaluated using a Jamar digital hand dynamometer (JLW Instruments, Chicago, IL, USA). Each participant performed three maximum grip attempts with both hands, with the arm fully extended at the side while standing. The highest value from each hand was averaged and used for subsequent analysis. All anthropometric and body composition assessments adhered to the guidelines established by ISAK (Marfell, 2012) [16,17].

### 2.5. Metabolic Risk Scoring System

A metabolic risk scoring system was developed by integrating body fat percentage (%BF), waist circumference (WC), and grip strength (GS). WC cut-off points were based on WHO guidelines: 80 cm and 88 cm for females and 94 cm and 102 cm for males. The %BF cut-offs were set at 30.0% and 35.0% for females and 20.0% and 25.0% for males.

GS thresholds were derived from Dodds et al. [18], categorizing weak strength below the 10th percentile and strong strength above the 90th percentile.

Participants were classified into four phenotypic risk groups based on their scores for %BF, WC, and GS (Table 1).

The total metabolic risk score was calculated by summing the scores for %BF, WC, and GS. Individuals were classified into four risk categories: protective state (score of 3), no risk (score of 4), increased risk (score of 5–7), and high risk (score of 8–9).

### 2.6. Evaluation of the Predictive Value of a Metabolic Risk Scoring System for Metabolic Syndrome

To evaluate the predictive capacity of the proposed metabolic risk classification system, a retrospective cohort of 166 clinical records of Mexican adults aged 18–65 years was analyzed. Inclusion criteria required complete records with anthropometric data, bioimpedance analysis, GS measurements, and laboratory results for fasting glucose, triglycerides, HDL cholesterol, and blood pressure. Data were obtained from routine medical check-ups conducted between January 2023 and October 2024.

Grip strength thresholds were determined using Dodds et al.’s criteria [18], with weak strength defined below the 10th percentile and strong strength above the 90th percentile. Metabolic syndrome was diagnosed following the Adult Treatment Panel III (ATP III) criteria, requiring at least three of the following: (1) abdominal obesity (WC ≥ 88 cm for females, ≥102 cm for males), (2) elevated triglycerides (≥150 mg/dL or on lipid-lowering therapy), (3) reduced HDL cholesterol (<50 mg/dL for females, <40 mg/dL for males), (4) elevated blood pressure (systolic ≥ 130 mmHg, diastolic ≥ 85 mmHg, or on antihypertensive therapy), and (5) elevated fasting glucose (≥100 mg/dL or on glucose-lowering therapy).

### 2.7. Statical Analyses

Statistical analyses were conducted using SPSS version 27 (IBM Corp., Armonk, NY, USA), and GraphPad Prism 8 (GraphPad Software, La Jolla, CA, USA). Data are presented as the mean ± standard deviation (SD). Group differences were evaluated using unpaired Student’s *t*-tests for normally distributed variables and the Kruskal–Wallis test for non-normally distributed variables, followed by Tukey’s post hoc test where applicable. Relationships between variables were assessed using Spearman’s rank correlation. The chi-square test and chi-square test for trend were used to compare categorical variables and assess associations between ordinal variables. A *p*-value of <0.05 was considered statistically significant.

## 3. Results

### 3.1. Anthropometric Characteristics

The study sample included 300 participants with an average age of 20.4 ± 1.6 years, 72% of whom were female. Detailed anthropometric data are provided in Table A1. Participants were categorized based on BMI: 3.7% were underweight, 55% had normal weight, 28.7% were people with overweight, and 12.7% were people with obesity. There were no significant differences in age or BMI between male and female participants (*p* = 0.112 and *p* = 0.052, respectively). The prevalence of people with overweight and obesity was higher in men (51.2%) than in women (37.5%). WC measurements indicated that 69.9% of females and 82.1% of males had WC values within the normal range per WHO guidelines. However, 50% of participants exceeded the threshold for high body fat percentage (>35% for females and >25% for males).

### 3.2. Distribution of Participants by %BF and WC

The distribution of the 300 participants according to the combination of %BF and WC, segregated by sex, is shown in Figure 1A. The number of subjects in the nine groups is detailed in Figure 1B. Among females, 48 (22.2%) were classified in the green group (absence of risk), 54 (25%) in the yellow group (slightly increased risk), 53 (24.5%) in the orange group (increased risk), 27 (12.5%) in the dark orange group (high risk), and 34 (15.7%) in the red group (very high risk). Among males, 25 (29.7%) were classified in the green group, 16 (19%) in the yellow group, 29 (34.5%) in the orange group, 7 (8.3%) in the dark orange group, and 7 (8.3%) in the red group.

### 3.3. Grip Strength and Correlations with Anthropometric Measures

GS averaged 26.28 ± 8.78 kg across the sample, with males demonstrating significantly greater strength (36.52 ± 8.61 kg) compared with females (22.29 ± 7.54 kg). According to Dodds et al. (2014) [18], 47.3% of participants were categorized as weak, with no significant sex-based differences (*p* = 0.525). Spearman’s correlation analysis revealed a strong negative correlation between body fat percentage and muscle percentage in both females (r = −0.97, *p* < 0.001) and males (r = −0.99, *p* < 0.001) (Figure A1). No significant correlation was observed between muscle percentage and GS in females (r = −0.02, *p* = 0.76), while a weak yet significant positive correlation was found in males (r = 0.30, *p* < 0.001) (Figure A2A,B). The correlation between body fat percentage and GS was positive but non-significant in females (r = 0.10, *p* = 0.12) and approached significance in a negative direction in males (r = −0.19, *p* = 0.06) (Figure A2C,D). A significant but weak positive correlation was observed between WC and GS in females (r = 0.30, *p* < 0.01), but no significant correlation was found in males (r = −0.01, *p* = 0.98) (Figure A2E,F).

### 3.4. Grip Strength Variation Across Metabolic Risk Categories

Participants were stratified into metabolic risk categories based on %BF, WC, and GS (Figure 2). This stratification utilized the color-coded risk phenotype classification, which now includes muscle strength as a critical factor. Table 2 provides a detailed breakdown of participants’ characteristics across the metabolic risk groups. Among females, significant differences in GS were observed across the risk categories. Specifically, the very high-risk group exhibited significantly greater GS compared with the no risk, slightly increased risk, and increased risk groups (*p* < 0.01). This indicates heterogeneity in muscle strength within these categories.

In contrast, no significant differences in GS were observed among males across the risk categories (*p* > 0.67), indicating a more uniform distribution of muscle strength across different risk phenotypes. However, within each risk category, there was notable variability in GS, with some individuals falling into the normal strength range, while others were classified as weak. To further explore this, a test of homogeneity was conducted to evaluate whether the distribution of muscle strength (weak, normal, strong) within each risk category and sex was homogeneous. The results showed that the differences in muscle strength distribution were highly significant across all sex and risk combinations (*p* < 0.0001), indicating that these groups are not homogeneous in terms of muscle strength. This finding suggests that measuring muscle strength is essential, as its variability within each risk category may provide crucial information for more precise stratification and clinical management of patients. Thus, muscle strength emerges as a mandatory parameter for a more accurate and personalized risk assessment.

### 3.5. Development and Implementation of a New Metabolic Risk Classification

Given the observed variability in muscle strength across the metabolic risk categories, we developed a new classification system that places muscle strength at its core (Figure 3). This system is simplified into a traffic light scoring model with four distinct categories: protective condition, no risk, increased risk, and high risk. Participants are assigned individual scores based on body fat percentage (BF), waist circumference (WC), and grip strength (GS). The total metabolic risk score is obtained by summing these individual scores, with higher totals indicating greater metabolic risk. The “protective condition,” represented in purple, captures individuals with a unique profile of low body fat but high muscle strength. This scoring system provides a clear and intuitive method to classify metabolic risk, with each category being distinctly represented by a color for easier interpretation.

### 3.6. Comparison of Phenotyping Systems for People with Obesity Incorporating Muscle Strength

Figure 4 compares the established phenotyping system for people with obesity based on %BF and WC, as described by Gomez-Ambrosi et al. [8], with the updated system that incorporates GS. The %BF–WC system identified 29.8% of males and 22.2% of females as having “no risk”, while 15.7% of females and 8.3% of males were categorized as “very high risk”. By incorporating GS into this phenotyping system, a significant redistribution across risk categories was observed. The proportion of participants classified as “no risk” decreased to 15.5% for males and 11.6% for females, demonstrating the added value of GS in risk assessment. The “increased risk” category expanded to include 69.0% of females and 70.2% of males, emphasizing the importance of muscle strength in refining risk classification. Additionally, a “protective condition” group emerged, comprising 2.4% of males, which was not identified by the original %BF–WC system. This redistribution highlights the critical role of muscle strength in metabolic risk assessment, providing a more comprehensive understanding of metabolic health and enabling the identification of individuals who might otherwise be overlooked.

### 3.7. Validation of the New Classification System in a Broader Population

To validate the new classification system, we analyzed 166 clinical records of Mexican adults aged 18–65 years. These records included anthropometric data, bioimpedance measurements, grip strength, and laboratory results for fasting glucose, triglycerides, HDL cholesterol, and blood pressure. Metabolic syndrome, defined by ATP III criteria, was assessed across the four risk categories (Figure 5A).

The retrospective cohort included 53.6% female and 46.4% male participants, which, while relatively balanced, differs from the cross-sectional dataset. Given the broader age range of this cohort compared with the cross-sectional dataset, we performed statistical adjustments to account for potential confounding effects. Multivariate regression analyses were conducted with age as a covariate, and additional stratified analyses by age groups (<30, 30–50, and >50 years) were performed to evaluate the consistency of the classification system across different age ranges. The results indicated that the metabolic risk classification remained valid across these subgroups. Given this consistency, we proceeded to analyze the overall dataset to evaluate the global applicability of the classification system.

In the protective and no risk groups, no cases of metabolic syndrome were observed (0/9 and 0/37, respectively). In contrast, the increased risk group showed a prevalence of 31.6% (18/57), and the high-risk group had a prevalence of 93.7% (59/63). The high-risk group demonstrated significantly higher odds for metabolic syndrome (OR = 28.23, 95% CI: 10.83–73.60, *p* < 0.001) compared with the increased risk group.

An analysis of metabolic parameters across categories (Figure 5B–F) revealed significant progressive deterioration with increasing risk levels (*p* < 0.001). Elevated fasting glucose, triglycerides, and blood pressure were prominent in the high risk group, while HDL cholesterol declined significantly. These findings validate the classification system’s ability to stratify individuals by metabolic health and highlight its utility in identifying those requiring targeted intervention.

Significant differences across risk categories were observed in weight, body mass index (BMI), body fat percentage, body fat mass, muscle mass percentage, muscle mass, visceral fat, waist circumference, hip circumference, and grip strength (*p* < 0.05). The data emphasize the physiological differences associated with varying risk levels. Statistical significance was determined using the ANOVA test for normally distributed variables and the Kruskal–Wallis test for non-normally distributed variables, with *p* < 0.05 being considered significant.

## 4. Discussion

This study introduces a novel metabolic risk classification system that integrates body fat percentage (%BF), waist circumference (WC), and muscle strength (grip strength, GS), providing a more comprehensive assessment of metabolic risk. Our findings, validated in a broader population of Mexican adults aged 18–65 years, underscore the importance of incorporating multiple dimensions of body composition into metabolic risk assessments. Muscle strength emerged as a critical yet often overlooked factor, demonstrating its ability to refine risk categorization and identify individuals with a heightened prevalence of metabolic syndrome.

A key finding of this study is the substantial reclassification of participants when muscle strength is included in the risk assessment. Previous phenotyping systems, such as the one proposed by Gomez-Ambrosi et al. (2023), rely solely on %BF and WC, categorizing a larger proportion of individuals as “no risk” [6]. However, our classification system reduces the proportion in the “no risk” category and increases those classified as increased or high risk. This shift demonstrates that conventional systems may underestimate metabolic risk, particularly in individuals with lower muscle strength, thereby missing opportunities for early intervention. The validation results further reinforce this, showing a prevalence of metabolic syndrome of 31.6% and 93.7% in the increased and high-risk categories, respectively, while no cases were observed in the protective or no-risk groups. These findings confirm the ability of the new classification system to identify at-risk individuals who might otherwise be overlooked by traditional methods, emphasizing its potential utility in guiding more targeted preventive strategies.

The identification of muscle strength as a critical factor in our new metabolic risk classification aligns with growing evidence that handgrip strength (HGS) serves as a reliable biomarker of overall health [13,19,20,21]. Just as HGS has been shown to predict various health outcomes, including cardiovascular disease and mortality, our findings suggest that integrating muscle strength into metabolic risk assessments offers a more comprehensive understanding of metabolic health [22,23]. This is particularly relevant in young adults, where traditional assessments may underestimate risk. The validation results reinforce this notion, showing a clear gradient in the prevalence of metabolic syndrome across risk categories, with no cases observed in the protective and no-risk groups, which exhibited higher muscle strength. By focusing on muscle strength, we can identify individuals at risk who might otherwise be overlooked by traditional methods, emphasizing the importance of early interventions aimed at enhancing muscle strength as a preventive measure against metabolic diseases [24].

This study also identifies a “protective condition” group—characterized by high muscle strength, normal %BF, and WC—introducing a crucial perspective in metabolic risk assessment. The lean and strong phenotype, theoretically offering protection against cardiometabolic diseases, was notably rare in our sample, with no females and only 2.4% of males (two individuals) meeting these criteria. In the validation cohort, this group similarly exhibited no cases of metabolic syndrome, further confirming its status as a truly low-risk category. This rarity underscores the importance of identifying not only those at high risk but also those with protective traits that may serve as benchmarks for optimal metabolic health. Furthermore, these findings challenge the assumption that leanness automatically equates to health, as individuals with normal %BF and WC but low muscle strength may still face significant metabolic risks. Therefore, strategies to promote muscle strength should not be limited to populations traditionally classified as high risk but should also extend to those presumed to be metabolically healthy based on conventional criteria, thereby ensuring a more comprehensive approach to prevention.

The absence of this protective phenotype in most participants underscores a critical public health issue: the need to move beyond conventional metrics of health that focus solely on body fat and waist circumference. Instead, our findings strongly advocate for the inclusion of muscle strength in health interventions, particularly for young adults. The validation results further highlight this necessity, revealing that individuals with lower muscle strength were disproportionately represented in the increased and high-risk categories, where metabolic syndrome prevalence reached 31.6% and 93.7%, respectively. These findings emphasize the urgent need for tailored strategies that address muscle strength deficits as a key component of metabolic risk reduction. Strength training, known to enhance insulin sensitivity and promote the secretion of health-protective myokines like exerkines, should be prioritized in public health initiatives to mitigate the growing burden of metabolic diseases [25,26,27].

Additionally, adequate protein intake combined with resistance training has been recognized as a fundamental strategy for maintaining optimal muscle function and reducing metabolic risk, particularly with aging [28]. Given that low muscle mass is a modifiable risk factor, interventions promoting increased physical activity—especially strength-based exercises—should be actively integrated into metabolic health guidelines.

This novel classification system not only uncovers hidden risks among individuals traditionally considered low risk but also provides a more nuanced approach to metabolic health. The validation results underscore its ability to accurately reclassify individuals into higher-risk categories, as evidenced by the significant increase in metabolic syndrome prevalence in the “increased risk” and “high risk” groups. By incorporating muscle strength as a central factor, this system offers a comprehensive framework for understanding metabolic well-being and promoting interventions that balance leanness and strength as dual pillars of cardiometabolic disease prevention.

Despite these strengths, several limitations must be acknowledged. The cross-sectional design of our study limits the ability to establish causality or determine the directionality of the observed relationships. Future longitudinal studies are needed to clarify whether low muscle strength is a precursor to metabolic risk or a consequence of underlying metabolic dysfunctions. Additionally, while the validation cohort expanded the applicability of the classification system to a broader age range (18–65 years), prospective studies will be essential to confirm these findings and evaluate the long-term predictive accuracy of the system across different life stages.

Moreover, the use of bioelectrical impedance analysis (BIA) for body composition measurement, although practical and widely accessible, has limitations in accuracy, particularly in individuals with extreme body compositions or fluid imbalances. Employing more precise methods, such as dual-energy X-ray absorptiometry (DXA), could strengthen the validity of future studies. Furthermore, integrating additional biomarkers—such as inflammatory markers, insulin resistance indices, or adipokine profiles—into the classification system could enhance its predictive capacity and provide deeper insights into the underlying mechanisms linking muscle strength to metabolic health.

Nevertheless, this study provides a robust, well-characterized sample and employs standardized protocols for anthropometric measurements and GS assessment. By integrating %BF, WC, and GS, the classification system offers a more nuanced understanding of metabolic risk, addressing gaps in conventional assessments. The validation results further underscore the system’s potential utility across a broader demographic, suggesting its relevance not only for young adults but also for middle-aged and older populations.

Looking ahead, validating this new classification system in diverse populations—including varying ethnicities, socio-economic backgrounds, age groups, and physical activity levels—will be crucial for its broader adoption and utility. Future research should also explore the longitudinal impact of muscle strength on metabolic health outcomes and evaluate whether interventions aimed at improving muscle strength can effectively reduce metabolic risk and improve long-term health outcomes. Such investigations are essential for establishing muscle strength as a cornerstone of metabolic risk assessment and management strategies.

## 5. Conclusions

This study presents a novel metabolic risk classification system integrating body fat percentage, waist circumference, and muscle strength, offering a more precise assessment of metabolic risk. The system effectively reclassifies individuals into appropriate risk categories, as demonstrated by its validation, highlighting muscle strength as a critical yet underutilized metric in metabolic health assessments.

We encourage further research to validate this classification system in different populations and settings. Incorporating muscle strength into routine risk assessments could enhance the identification of at-risk individuals and support the development of more effective prevention strategies for metabolic diseases.

## Figures and Tables

**Figure 1 jfmk-10-00072-f001:**
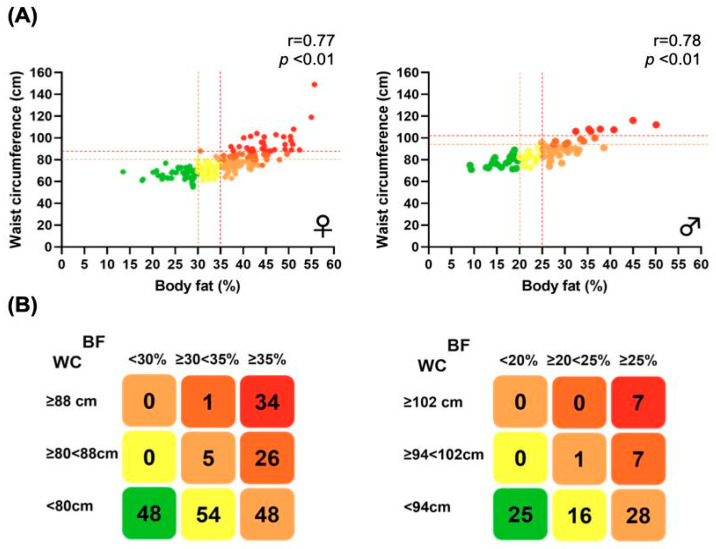
Distribution of participants by %BF and WC. (**A**) Graphical distribution of %BF and WC in 216 females (**left**) and 84 males (**right**). Vertical dashed lines indicate cut-offs for defining overweight and obesity according to %BF (30.0% and 35.0% in females and 20.0% and 25.0% in males, respectively), while horizontal lines indicate cut-offs for defining increased or high cardiometabolic risk (80 and 88 cm in females and 94 and 102 cm in males, respectively) according to WC. Colors denote the five different phenotypes according to cardiometabolic risk: green (no risk), yellow (slightly increased risk), orange (increased risk), dark orange (high risk), and red (very high risk). Spearman’s correlation was used to assess the relationship between %BF and WC. (**B**) Matrix classification according to %BF and WC cut-off points for females (**left**) and males (**right**), defining the five different risk phenotypes. The number of subjects in each of the nine squares is indicated. %BF, body fat percentage; WC, waist circumference.

**Figure 2 jfmk-10-00072-f002:**
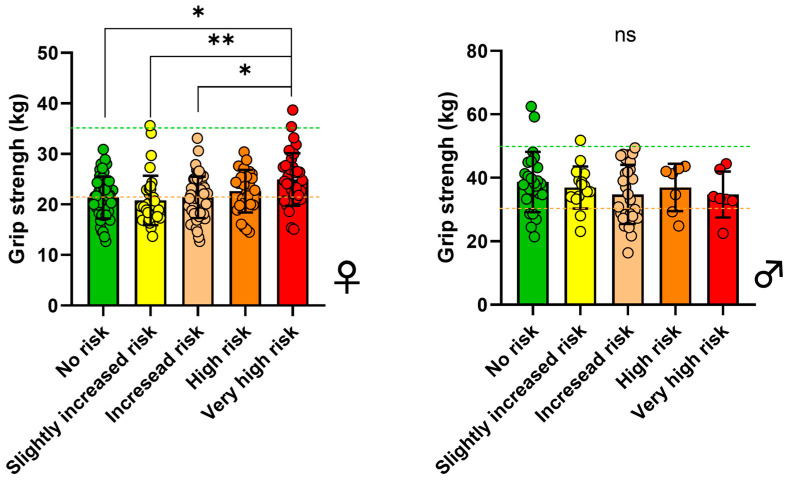
Maximum GS by metabolic risk categories. The number of female subjects (**left**) in each group was 48, 54, 53, 27, and 34. The number of male subjects (**right**) in each group was 25, 16, 29, 7, and 7. Horizontal dashed lines indicate cut-offs for defining weak, normal, and strong according to Dodds et al., 2014, [18], with weak defined as <21 kg, normal as 21–35 kg, and strong as >35 kg in females; and weak as <35 kg, normal as 36–52 kg, and strong as >52 kg in males. Bars represent the mean ± SD. Statistical differences between groups were analyzed by an ANOVA test. * *p* < 0.05 and ** *p* < 0.01, and ‘ns’ indicates non-significant differences.

**Figure 3 jfmk-10-00072-f003:**
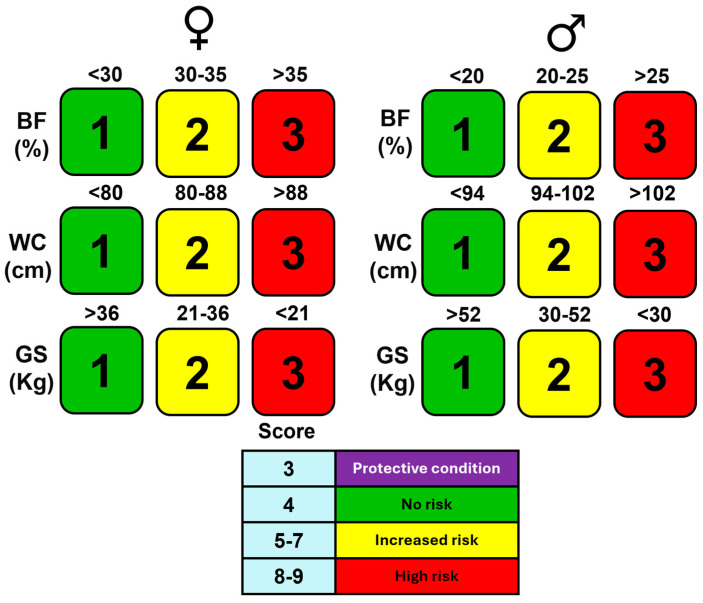
Metabolic risk scoring system by sex using %BF, WC, and GS. This scoring system categorizes cardiometabolic risk into four distinct levels: purple (protective condition), green (no risk), yellow (increased risk), and red (high risk). Each participant is assigned a score based on body fat percentage (BF), waist circumference (WC), and grip strength (GS). The total metabolic risk score is obtained by summing the scores from each of these three categories. Lower total scores indicate a protective condition, while higher scores correspond to increased and high risks. The “protective condition”, represented in purple, is particularly relevant for individuals with low body fat but high muscle strength, highlighting a unique metabolic profile. This traffic light system simplifies the identification of metabolic risk, enhancing both clinical interpretation and patient communication.

**Figure 4 jfmk-10-00072-f004:**
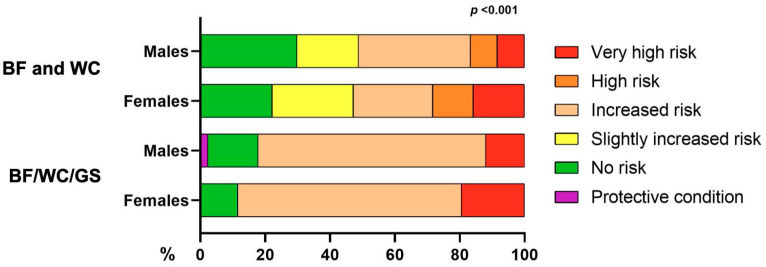
Comparison of %BF–WC and %BF–WC–GS scoring systems. Differences in distribution were calculated using the chi-square test.

**Figure 5 jfmk-10-00072-f005:**
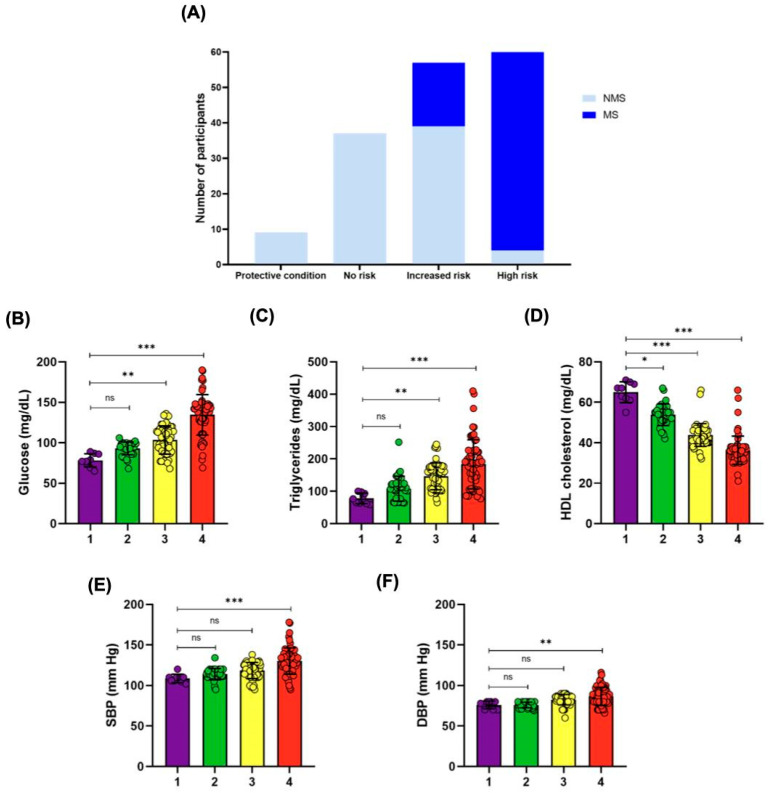
Metabolic Parameters Across Risk Categories in the Validation Cohort. (**A**) Prevalence of metabolic syndrome (MS, blue) and non-metabolic syndrome (NMS, light blue) participants across the four metabolic risk categories: protective condition (1), no risk (2), increased risk (3), and high risk (4). (**B**) Fasting glucose levels (mg/dL) significantly increase across risk categories. (**C**) Triglyceride levels (mg/dL) show a significant rise with higher metabolic risk. (**D**) High-density lipoprotein (HDL) cholesterol levels (mg/dL) decrease progressively across the categories. (**E**) Systolic blood pressure (SBP, mmHg) and (**F**) diastolic blood pressure (DBP, mmHg) increase significantly in higher-risk groups. Statistical significance is denoted as follows: ns, not significant; * *p* < 0.05; ** *p* < 0.01; *** *p* < 0.001. Data are presented as the mean ± standard error. Statistical analyses were conducted using a one-way ANOVA test followed by Tukey’s post hoc test.

**Table 1 jfmk-10-00072-t001:** Classification criteria for metabolic risk based on body fat percentage, waist circumference, and grip strength.

Category	Points	%BF Cut-Offs	WC Cut-Offs (cm)	GS Cut-Offs (kg)
Women	1	<30%	<80 cm	>36 kg (strong)
	2	30–35%	80–88 cm	21–36 kg (moderate)
	3	≥35%	≥88 cm	<21 kg (weak)
Men	1	<20%	<94 cm	>52 kg (strong)
	2	20–25%	94–102 cm	30–52 kg (moderate)
	3	≥25%	≥102 cm	<30 kg (weak)

**Table 2 jfmk-10-00072-t002:** Characteristics of participants stratified by risk categories.

Parameter	No Risk (*n* = 73)	Slightly Increased Risk (*n* = 70)	Increased Risk (*n* = 82)	High Risk (*n* = 34)	Very High Risk (*n* = 41)	*p*
Age (years)						
All subjects	20.18 ± 1.55	20.33 ± 1.44	20.79 ± 2.01	20.47 ± 1.86	20.46 ± 1.45	0.234
Males	20.68 ± 1.65	21 ± 2.17	20.93 ± 2.92	20 ± 0.81	20.29 ± 0.95	0.02
Females	19.92 ± 1.45	20.11 ± 1.07	20.7 ± 1.3	20.5 ± 2.04	20.5 ± 1.54	0.747
Weight (kg)						
All subjects	57 ± 11	59.92 ± 10.1	66.24 ± 10.2	73.69 ± 10.68	88.81 ± 4.67	<0.001
Males	68.74 ± 8.59	73.6 ± 6.46	75.4 ± 8.94	91.35 ± 7.13	99.38 ± 10.32	<0.001
Females	50.9 ± 6.09	55.87 ± 7.03	61.2 ± 6.95	69.1 ± 5.23	86.63 ± 14.59	<0.001
Height (cm)						
All subjects	164.9 ± 9	163 ± 9.1	162 ± 8.7	162 ± 9.27	163 ± 7.65	0.566
Males	174 ± 6.2	174.5 ± 5.16	170 ± 8.04	177 ± 7.08	171 ± 6.4	0.301
Females	160 ± 5.95	159 ± 7.05	158 ± 6.05	158 ± 4.9	161 ± 6.7	0.186
Body Mass Index (kg/m^2^)						
All subjects	20.7 ± 2.31	22.38 ± 2.08	24.84 ± 2.02	27.82 ± 2.14	33.24 ± 5.04	<0.001
Males	22.6 ± 2.15	24.1 ± 1.41	25.94 ± 1.81	29.12 ± 1.79	33.64 ± 2.89	
Females	19.8 ± 1.77	21.8 ± 1.97	24.23 ± 1.88	27.49 ± 2.12	33.1 ± 5.4	
Body Fat (%)						
All subjects	22.5 ± 5.67	29.88 ± 4.31	34.72 ± 5.26	39.41 ± 5.86	44.56 ± 5.42	<0.001
Males	16.2 ± 3.12	22.48 ± 1.59	28.71 ± 3.12	31.48 ± 3.35	39.58 ± 6.18	<0.001
Females	25.7 ± 3.57	32 ± 1.43	38 ± 2.59	41.4 ± 4.45	45.5 ± 4.72	<0.001
Body Fat (kg)						
All subjects	12.4 ± 2.76	17.61 ± 2.43	22.77 ± 3.1	28.78 ± 4.34	39.84 ± 10.09	<0.001
Males	11.25 ± 2.86	16.57 ± 2.17	21.87 ± 3.2	28.76 ± 3.77	39.62 ± 9.27	<0.001
Females	13.1 ± 2.51	17.9 ± 2.43	23.2 ± 1.88	28.7 ± 4.54	39.88 ± 10.38	<0.001
Body Muscle (%)						
All subjects	42.3 ± 4.38	38.16 ± 3.23	35.72 ± 3.68	33.27 ± 3.74	30.66 ± 3.17	<0.001
Males	47.3 ± 1.92	43.7 ± 0.97	40 ± 2.02	38.85 ± 2.1	34 ± 3.89	<0.001
Females	39.7 ± 2.73	36.5 ± 1.11	33.34 ± 1.62	31.83 ± 2.49	29.95 ± 2.53	<0.001
Body Muscle (kg)						
All subjects	24.45 ± 6.75	23.12 ± 5.73	23.92 ± 5.83	24.73 ± 5.96	27.13 ± 4.69	0.015
Males	32.51 ± 4.17	32.17 ± 2.76	30.27 ± 4.32	35.5 ± 3.57	33.71 ± 4	<0.001
Females	20.2 ± 2.91	20.4 ± 2.93	20.45 ± 2.86	21.93 ± 1.69	25.77 ± 3.56	0.028
Visceral Fat (AU)						
All subjects	4.6 ± 1.3	7.2 ± 1.23	10.51 ± 2.08	13.76 ± 3.3	17.58 ± 2.32	<0.001
Males	4 ± 1.35	6.6 ± 1.08	8.8 ± 1.61	12.1 ± 1.86	17 ± 2.7	<0.001
Females	5 ± 1.13	7.3 ± 1.23	11.41 ± 1.73	14.1 ± 3.51	17.7 ± 2.26	<0.001
Waist Circumference (cm)						
All subjects	70.7 ± 7.2	73.27 ± 6.88	79.31 ± 7.45	86.15 ± 5.87	98.3 ± 11.84	<0.001
Males	78.3 ± 4.89	82.58 ± 5.03	87.35 ± 5.06	96.68 ± 2.26	109 ± 3.65	<0.001
Females	66.8 ± 4.63	70.5 ± 4.55	74.9 ± 4.11	83.42 ± 2.24	96.09 ± 11.74	<0.001
Hip Circumference (cm)						
All subjects	91 ± 5.65	93.87 ± 6.76	98.5 ± 6.19	104.32 ± 5.06	114.07 ± 11.66	<0.001
Males	94.7 ± 5.62	99 ± 3.61	100.5 ± 5.43	109.4 ± 4.75	114.1 ± 5.24	<0.001
Females	89 ± 4.63	92.3 ± 6.74	97.5 ± 6.36	102.9 ± 4.29	114 ± 12.64	<0.001
Maximum Grip Strength (kg)						
All subjects	27.3 ± 10.46	25.52 ± 8.24	26.13 ± 9.06	25.58 ± 7.66	26.61 ± 6.65	0.773
Males	38.6 ± 9.5	36.95 ± 6.6	34.76 ± 9.35	36.94 ± 7.47	34.74 ± 7.24	0.01
Females	21.3 ± 4.12	22.1 ± 4.98	21.41 ± 4.1	22.64 ± 4.23	24.94 ± 5.21	0.671
Physical Activity (%)						
All subjects	60.3	68.6	64.6	58.8	46.3	0.206
Males	88	93.8	69	85.7	42.9	0.339
Females	45.8	61.1	62.3	51.9	47.1	0.031
Smoking (%)						
All subjects	32.9	31.4	30.5	41.2	29.3	0.82
Males	40	25	27.6	42.9	28.6	0.87
Females	29.2	33.3	32.1	40.7	29.3	0.778
Alcohol Use (%)						
All subjects	86.3	90	85.4	70.6	90.2	0.088
Males	92	81.3	86.2	71.4	100	0.115
Females	83.3	92.6	84.9	70.4	88.2	0.469

## Data Availability

Data will be made available upon reasonable request by accredited researchers.

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
