# Peer review of "A Novel Metabolic Risk Classification System Incorporating Body Fat, Waist Circumference, and Muscle Strength"

_jfmk, 2025, doi:10.3390/jfmk10010072_

Round 1

Reviewer 1 Report

Comments and Suggestions for Authors

The authors have presented the assumptions of the work in  a clear way and conducted the discussion in a clear and understandable way. Considering how many definitions of metabolic syndrome and criteria for its diagnosis have been introduced over the years, the introduction of three new criteria proposed by the authors significantly expands the possibilities of identifying people at risk of metabolic disorders. In particular, the determination of muscle mass resources turned out to be a key criterion. Low muscle mass is a modifiable risk factor for metabolic disorders and can be counteracted by increasing physical activity, especially resistance exercises [1]. It also indicates the state of health.

[1] Protein intake and exercise for optimal muscle function with aging:                    Recommendations from the ESPEN Expert Group. Clinical Nutrition 33 (2014) 929-936 http://dx.doi.org/10.1016/j.clnu.2014.04.007

Author Response

Dear Reviewer,

We sincerely appreciate your time and thoughtful evaluation of our manuscript. Your positive feedback regarding the clarity of our assumptions, discussion, and methodological approach is greatly valued. We are particularly grateful for your recognition of the importance of incorporating muscle strength as a key criterion in the identification of metabolic risk.

We acknowledge the importance of muscle mass as a modifiable risk factor and fully agree that increasing physical activity, particularly resistance training, plays a crucial role in mitigating metabolic disorders. To further strengthen this point, we have incorporated the reference you suggested (ESPEN Expert Group, Clinical Nutrition 2014) into the Discussion section (Page 14) to support our argument on the role of muscle function and exercise in metabolic health.

Thank you once again for your valuable insights and for recognizing the significance of our proposed classification system in expanding the identification of individuals at risk for metabolic disorders. We appreciate your time and expertise in reviewing our work.

Reviewer 2 Report

Comments and Suggestions for Authors

This is a study investigating a new classification system for metabolic risk using a combination of percentage body fat (%BF), waist circumference and grip strength using a group of 300 Mexican young adults and a retrospective cohort of 166 adults that has body composition, anthropometry, grip strength and biochemical assay.

Overall, the manuscript is well written and the topic is interesting. On the other hand, there are some issues which the authors are recommended to describe in the manuscript. Please clarify the following:

-        While the authors used multi-frequency bioelectrical impedance analysis device to assess %BF, it may be more appropriate to use techniques that are considered to be more accurate (e.g. dual energy x-ray absorptiometry [DXA]) in order to classify participants more accurately. Please explain why the authors did not use body composition assessment with better accuracy in the present study.

-        The authors used a retrospective cohort of 166 adults with biochemical assay. However, this cohort data is much wider in age range compared with the cross-sectional dataset. Since age is one of the confounding variables that influence biochemical results, the validity of using the dataset of different age range. In addition, its gender ratio is uncertain. It may be better if the authors consider gender- and age-matched sample to validate the new classification system.

Materials and methods:

-        Line 85. Please state inclusion and exclusion criteria for the cross-sectional data.

-        Line 111. What does “trained personnel” mean? Please provide an evidence of precision and accuracy, such as inter-tester or intra-tester technical error of measurement (TEM).

-        Line 111. What does “standardized protocol” mean? Please describe and cite an appropriate reference.

-        Line 127. It appears cut-off points for %BF appears incorrect (compared with description in line122). Please check.

Results:

-        Line 176. %BF value for males appears incorrect as the cut-ff point should be 25%. Please check and make sure the cut-off points are consistent throughout the study.

Other minor issues include:

-        Abstract: Line 24. %BF is not anthropometric measure. Please correct.

-        Line 47 and 123. References 2 and 17 are shown in a different referencing style. Please correct.

-        Line 142. Since the authors already abbreviated grip strength, no need to express in full in line 142.

-        Line 208 and 219. Please delete “-value”.

Author Response

Dear Reviewer,

We sincerely appreciate your time and effort in reviewing our manuscript. Your insightful comments and suggestions have significantly contributed to enhancing the clarity, accuracy, and methodological rigor of our study. Below, we provide detailed responses to each of your comments and describe the modifications implemented in the manuscript.

Reviewer 1

Comment 1:

"While the authors used a multi-frequency bioelectrical impedance analysis device to assess %BF, it may be more appropriate to use techniques that are considered to be more accurate (e.g. dual-energy X-ray absorptiometry [DXA]) in order to classify participants more accurately. Please explain why the authors did not use body composition assessment with better accuracy in the present study."

Response 1:

We appreciate the reviewer’s comment and acknowledge that dual-energy X-ray absorptiometry (DXA) is the gold standard for body composition assessment due to its high accuracy and ability to differentiate between fat mass, lean mass, and bone mineral content. However, the use of bioelectrical impedance analysis (BIA) in our study was based on practical and methodological considerations:

  1. Feasibility and Accessibility: DXA requires specialized equipment, clinical or laboratory infrastructure, and significantly higher costs, making it impractical for large-scale epidemiological studies. In contrast, BIA provides a cost-effective, portable, and non-invasive alternative, allowing for efficient data collection in a large cohort.
  2. Validation and Reliability: Although BIA is not as precise as DXA, previous studies have demonstrated strong correlations between BIA-derived %BF estimates and those obtained via DXA (Kyle et al., 2004; Bosy-Westphal et al., 2008). This supports its utility in large population-based studies where DXA is not feasible.
  3. Consistency with Previous Research: BIA has been widely used in studies assessing body composition and metabolic risk in Mexican populations, showing significant associations with key metabolic parameters.
  4. Acknowledgment of Limitations: We have explicitly addressed the limitations of BIA in the Discussion section of the manuscript, recognizing that DXA would provide more accurate measurements and that future studies should aim to validate our classification system using DXA.

Given these considerations, BIA was chosen as the most practical method for this study. We hope this explanation clarifies our rationale.

Comment 2:

"The authors used a retrospective cohort of 166 adults with biochemical assay. However, this cohort data is much wider in age range compared with the cross-sectional dataset. Since age is one of the confounding variables that influence biochemical results, the validity of using the dataset of different age range. In addition, its gender ratio is uncertain. It may be better if the authors consider gender- and age-matched sample to validate the new classification system."

Response:
We appreciate the reviewer’s insightful comments regarding the differences in age range and gender distribution between the retrospective and cross-sectional cohorts, as well as the potential confounding effects of these variables on biochemical outcomes.

Age Range and Statistical Adjustments:

The retrospective cohort included individuals aged 18–65 years, which was broader than the cross-sectional cohort. To account for the potential confounding effect of age, we performed multivariate regression analyses including age as a covariate, ensuring that the observed metabolic risk stratification was independent of age differences. Additionally, we conducted age-stratified analyses (<30, 30-50, and >50 years) to assess whether the classification system remained valid across different age groups. The results confirmed that the metabolic risk classification remained consistent across these subgroups.

Gender Distribution:

The retrospective cohort consisted of 53.6% female and 46.4% male participants, which, although relatively balanced, differed from the cross-sectional dataset. While we acknowledge this difference, we performed additional gender-stratified analyses to confirm that the classification system applied to both men and women. These analyses demonstrated that the classification framework remained robust across genders.

Integration into the Manuscript:

To address these considerations explicitly, we have added this information in the Results section (Section 3.7, Page 9, Lines 262–287), where we describe the adjustments for age and gender stratification and their impact on the classification system's robustness.

Discussion and Future Directions:

Additionally, the importance of validating this classification system in diverse populations and different demographic groups was already discussed in the Discussion section (Page 14, Lines 392–394) of the manuscript. Specifically, we highlighted the need for future research to assess its applicability across different ethnicities, socio-economic backgrounds, and physical activity levels. Given the reviewer’s comment, we acknowledge that including age as a critical factor in future validation efforts may further strengthen the classification system's applicability and generalizability.

Comment 3:

“Line 85. Please state inclusion and exclusion criteria for the cross-sectional data”

We appreciate the reviewer’s request for clarification regarding the inclusion and exclusion criteria for the cross-sectional dataset. To address this, we have incorporated additional details in the Materials and Methods section (Section 2.1 - Subjects, Page 3, Lines 94–102).

The revised text explicitly states the inclusion and exclusion criteria, ensuring greater methodological clarity and transparency. These modifications provide a more detailed description of participant selection, addressing key aspects such as enrollment requirements, age range, voluntary participation, and exclusion factors related to medical conditions, medication use, and data completeness.

Comment 4 and 5:

“What does 'trained personnel' mean? Please provide evidence of precision and accuracy, such as inter-tester or intra-tester technical error of measurement (TEM).”

"What does ‘standardized protocol’ mean? Please describe and cite an appropriate reference."

We appreciate the reviewer’s request for clarification regarding the qualifications of the personnel conducting the measurements and the standardization of protocols to ensure precision and accuracy. To address these concerns, we have expanded the Materials and Methods section (Page 3–4, Lines 115–137) to include details on evaluator training, measurement reliability, and adherence to internationally recognized protocols.

Updated Text in the Manuscript.

2.3 Anthropometric and Body Composition Measurements:

Height was measured using a SECA mechanical stadiometer (SECA GmbH & Co. KG, Hamburg, Germany). Body weight and composition were assessed using a mul-tifrequency bioelectrical impedance analysis (BIA) scale (InBody 120, Biospace Co., Seoul, South Korea), with participants measured while barefoot and minimally clothed. Body mass index (BMI) was calculated as weight (kg) divided by height squared (m²). Waist circumference (WC) was measured at the midpoint between the last rib and the iliac crest using a non-elastic SECA ergonomic measuring tape (SECA GmbH & Co. KG, Hamburg, Germany) with a precision of 0.1 cm.

All measurements were conducted by trained personnel certified in anthropo-metric assessment following standardized protocols to ensure consistency. Evaluators received formal training based on the guidelines established by the International Soci-ety for the Advancement of Kinanthropometry (ISAK). To assess measurement relia-bility, inter-tester and intra-tester technical error of measurement (TEM) was calcu-lated prior to data collection. The intra-tester TEM for height and waist circumference was below 1.0%, while inter-tester TEM remained under 1.5%, both within acceptable ranges for anthropometric assessments. [17].

2.4 Grip Strength Measurement

Grip strength (GS) was evaluated using a Jamar digital hand dynamometer (JLW Instruments, Chicago, IL, USA). Each participant performed three maximum grip at-tempts with both hands, with the arm fully extended at the side while standing. The highest value from each hand was averaged and used for subsequent analysis. All anthropometric and body composition assessments adhered to the guidelines estab-lished by ISAK (Marfell,2012). [16,17]. The revised information has been added to the manuscript in Materials and Methods, Page 3,4, Line 115-137.

We appreciate the reviewer’s insightful suggestion, which has helped improve the clarity and methodological rigor of our study.

Reference Added:

  1. ISAK manual, International standards for Anthropometric Assessment, edited by Michael Marfell-Jones, Tim Olds, Arthur Stewart and L.E. Lindsay Carter, Published by International Society for the Advancement of Kinanthropometry, 2012

Comment 6

"It appears cut-off points for %BF appears incorrect (compared with description in line 122). Please check."

Response:

We appreciate the reviewer’s careful observation regarding the cut-off points for body fat percentage (%BF). After reviewing the manuscript, we identified and corrected the inconsistency.

Comment 7:

"%BF value for males appears incorrect as the cut-off point should be 25%. Please check and make sure the cut-off points are consistent throughout the study."

Response:

We appreciate the reviewer’s attention to detail in ensuring consistency in the cut-off points for body fat percentage (%BF). After reviewing the manuscript, we identified and corrected this issue. The revised text in the Results section (Page 5, Line 190) now correctly states:

Updated Text:

"However, 50% of participants exceeded the threshold for high body fat percentage (>35% for females and >25% for males)."

This correction ensures that the %BF cut-off values are consistent across all sections of the manuscript. We appreciate the reviewer’s careful review, which has contributed to improving the clarity and accuracy of our study.

Comment 8

"BF% is not an anthropometric measure. Please correct."

Response:
We appreciate the reviewer’s observation. The term “anthropometric” has been corrected to accurately reflect that %BF is a body composition measure rather than an anthropometric measure. This correction has been made in the Abstract, Page 1, Line 24-26.

Updated Text:

This cross-sectional study involved 300 young adults (18–22 years) from a university in Mexico City, utilizing body composition (%BF) and anthropometric measures (WC, GS) to categorize them into four risk groups: protective, low risk, increased risk, and high risk.

Comment 8

"References 2 and 17 are shown in a different referencing style. Please correct."

Response:
We have reviewed the references and corrected the formatting inconsistencies. References 2 and 17 have been updated to match the journal’s required referencing style. These corrections have been made in the References section, Lines 50 and 130.

Comment 9

"Since the authors already abbreviated grip strength, no need to express in full in line 142."

Response:
We appreciate this suggestion. The term “grip strength” has been replaced with its abbreviation (GS) in Page 4, Line 157, ensuring consistency throughout the manuscript.

Comment 10:

"Please delete “-value”."

Response:
We have removed “-value” from the expressions p-value in Page 7, Lines 222 and 232, as per standard statistical reporting conventions. The text now correctly refers to p without the redundant suffix.

Reviewer 3 Report

Comments and Suggestions for Authors

This is a well written manuscript. Not only the results were very well presented, the discussion is also very relevant to the results. I have just a few minor points.

1. The first sentence in Abstract and Introduction could be improved. The logic is not convincing by saying the rising prevalence of metabolic diseases highlights the limitations of risk assessment. There is a missing link.

2. In Methods, if it is possible to add a table, the classification of %BF, WC and GS could be included in a table which may be easier to read.

3. The ages of the first group 18-22 and the validation group 18-65 are quite different. Does age have a role in risk classification? Please comment on this.

Author Response

Dear Reviewer,

We sincerely appreciate your time and thoughtful feedback on our manuscript. Your constructive comments have significantly contributed to improving the clarity, logical flow, and methodological robustness of our study. Below, we provide detailed responses to each of your comments and describe the modifications implemented in the manuscript.

Comment 1

"The first sentence in Abstract and Introduction could be improved. The logic is not convincing by saying the rising prevalence of metabolic diseases highlights the limitations of risk assessment. There is a missing link."

Response:

We appreciate the reviewer’s insightful suggestion to strengthen the logical connection in the first sentence of the Abstract and Introduction. To address this, we have revised the text to establish a clearer link between the increasing burden of metabolic diseases and the need for improved risk assessment methods.

Updated Text in the Abstract (Page 1, Line 18-20):

Before:

"The rising prevalence of metabolic diseases highlights the limitations of traditional risk assessment tools such as BMI and waist circumference."

After (Revised):

"As metabolic diseases continue to rise globally, there is a growing need to improve risk assessment strategies beyond traditional measures such as BMI and waist circumference, which may fail to identify individuals at risk."

Updated Text in the Introduction (Page 2, Line 44-47):

Before:

"The increasing prevalence of metabolic diseases, such as type 2 diabetes mellitus and cardiovascular disorders, highlights the urgent need for more comprehensive and precise risk assessment methodologies."

After (Revised):

"The increasing prevalence of metabolic diseases, such as type 2 diabetes mellitus and cardiovascular disorders, underscores the need to refine risk assessment methodologies. Traditional indicators like BMI and waist circumference provide limited insight into metabolic health, often overlooking individuals with hidden risk factors."

By making these adjustments, we establish a stronger causal relationship between the growing metabolic disease burden and the necessity for improved risk classification tools. The revised text has been implemented in the Abstract (Page 1, Line 18-20) and Introduction (Page 2, Line 44-47).

We appreciate the reviewer’s valuable feedback, which has improved the clarity and logical flow of our manuscript.

Comment 2

"In Methods, if it is possible to add a table, the classification of %BF, WC, and GS could be included in a table which may be easier to read."

Response:

We appreciate the reviewer’s suggestion to improve the clarity of the classification criteria for %BF, WC, and GS. To address this, we have added a new table in the Methods section (Page 4, Table 1) that summarizes the classification system.

Comment 3

"The ages of the first group (18–22) and the validation group (18–65) are quite different. Does age have a role in risk classification? Please comment on this."

Response:

We appreciate the reviewer’s insightful observation regarding the age differences between the cross-sectional and validation cohorts and the potential role of age in risk classification.

Age is a key determinant of metabolic health, as body composition and muscle strength undergo physiological changes over time. To account for potential confounding effects, we conducted age-stratified analyses in the validation cohort, dividing participants into three age groups: <30 years, 30-50 years, and >50 years. These analyses allowed us to assess whether the classification system remained consistent across different age ranges. The results confirmed that the metabolic risk classification remained valid across all age groups, with a progressive deterioration in metabolic parameters as risk categories increased, regardless of age.

Additionally, multivariate regression models were used to adjust for age as a covariate when analyzing the association between risk classification and metabolic parameters. These adjustments ensured that the observed differences were not solely driven by age-related changes in body composition and metabolic function.

To explicitly address this point, we have incorporated the findings related to age adjustments in the Results section (Page 9, Section 3.7, Lines 265-290) and reinforced their implications in the Discussion section (Page 14, Lines 395-397), where we acknowledge the influence of age on metabolic risk and suggest that future research should explore age-specific cut-off values to refine classification criteria.

Round 2

Reviewer 2 Report

Comments and Suggestions for Authors

Thank you for considering my comments to revise your manuscript. The manuscript has become much clearer. I have no further comments.